# Proportional Reasoning Deficit in Dyslexia

**DOI:** 10.3390/brainsci13050795

**Published:** 2023-05-12

**Authors:** Giulia Cappagli, Beatrice Carzola, Cristina Potente, Monica Gori

**Affiliations:** 1Unit for Visually Impaired People, Istituto Italiano di Tecnologia, 16125 Genova, Italy; 2Centro Leonardo, 16129 Genova, Italy

**Keywords:** dyslexia, proportional reasoning, mathematical thinking

## Abstract

Dyslexia has been linked to an altered perception of metrical structures in language, but no study to date has explored the link between reading impairments and other forms of metrical thinking (e.g., proportional reasoning). In the present study, we assessed proportional reasoning in 16 dyslexic children and 16 age-matched controls from 7 to 10 years of age in order to investigate whether dyslexia might be also linked to an altered form of metrical thinking. We found that dyslexic children are less accurate in performing judgments about proportionality compared to typical peers and that reading accuracy correlates with proportional reasoning abilities for 7–8-year-old children. Overall, these findings suggest that a link exists between reading and proportional reasoning abilities. We might speculate that fostering reasoning based on the meter can facilitate reading because it permits the segmentation of words in syllables and that dyslexia can be identified early with alternative non-reading tasks such as the proportional reasoning task used in this work.

## 1. Introduction

Dyslexia is typically defined as a language-related impairment, dealing with the discrepancy between the expected reading performance based on general intelligence quantification and the actual reading performance [1]. Several works have shown that other perceptual or cognitive deficits may contribute to determining the behavioral performance of dyslexic individuals. These impairments are related to abnormal visual processing [2], impaired phonological abilities [3,4,5,6,7,8], or altered attentional recruitment [9,10]. Nonetheless, it is still unclear whether such deficits represent the core cause of dyslexia or, instead, the behavioral manifestation of a broader and more profound impairment. A line of evidence has recently shown that dyslexia symptoms are linked with altered rhythmic perception in the musical domain [11,12,13,14,15,16] and the language domain [17]. Specifically, the impairment of rhythmic abilities in the language domain has been related to an altered sound rise time perception [5,18], which might determine poor phonemic discrimination and, thus, reduced sensitivity to the metrical structure of language as well as altered phonological awareness, which is a typical precursor of reading skills [11,19]. While the pattern of stressed and unstressed syllables in a verse or a sentence represents the metrical structure of language, the pattern of beats in a composition represents the metrical structure of music. Indeed, both domains (language and music) shared a common structure based on “meter”, which arises from whole partitioning and could be represented by a syllable in the case of language and a note in the case of music. In the past decades, a substantive body of research has demonstrated that dyslexia symptoms are linked to altered auditory and visual processing [2,3,4,5,6,7,8] and that early mathematics skills are predictive of later reading achievement [20,21,22,23,24,25]. However, no study to date has directly linked reading impairments to a mathematical form of thinking that relies on a metrical structure, such as proportional reasoning. 

Proportional reasoning is a form of thinking based on the ability to relate parts in a whole and understand the multiplicative relationships between two co-varying measures [26]. Broadly, in a typical comparison problem performed to assess proportional reasoning abilities, participants are given four numbers (a, b, c, d), and a judgment is required about the values of the ratios in terms of whether or not they are equal, or which ratio is greater (e.g., whether a juice made of 2 red and 10 blackberries taste exactly the same as a juice made of 6 red and 30 blackberries). Most of the studies about proportional reasoning focused on investigating its developmental time course (e.g., [27,28,29,30,31,32]) as well as identifying problem characteristics that can influence the difficulty of proportional reasoning (e.g., the size of the two comparisons, see [33]). However, there is still an ongoing debate about how much the nature of the quantities included in the problem (e.g., whether the quantities presented in a comparison problem are made of discrete or continuous units) influences proportional reasoning abilities. Specifically, some authors have shown that the nature of these quantities has no influence on children’s performance levels [34], while others demonstrated that comparison problems involving continuous units are easier to solve than comparison problems containing discrete units [30,35,36,37,38]. This second line of evidence suggested that children’s difficulties with discrete units arise from their propensity to match the number of units in the comparison elements instead of focusing on the proportional relations between elements. This is coherent with what has been previously hypothesized about young children’s tendency to focus on the single parts of a proportion instead of the whole-part relation [39,40]. In other words, it seems that when typical children are wrong in proportional reasoning tasks, it is because they remained anchored to the segmented units of a whole, and this brings them to use visual comparison and counting strategies to get numerical instead of proportional equivalence. Overall, such results suggest that reading and mathematical abilities are strongly linked since they both rely on relative comparisons, with proportional reasoning being based on the ability to identify the relation between parts in a whole and reading is based on syllable segmentation which is crucial for the development of phonological awareness. In line with this, it has been shown that language abilities explain individual differences in mathematical ability [41,42,43,44,45], and language and patterning abilities explain individual differences in young children’s proportional reasoning abilities [46]. 

With the main aim to further demonstrate the strong link between language and mathematical domains, the present study aims to assess whether dyslexia symptoms relate to a more general deficit in the form of metrical thinking, e.g., proportional reasoning. Indeed, a general difficulty in relating parts in a whole might determine a difficulty in syllable segmentation leading to a reading impairment, as has already been shown by Swan et al. when demonstrating a relation between phonological awareness and reading impairments [47]. To test this hypothesis, we involved a group of sixteen 7-to-10 years old children with dyslexic symptoms and sixteen age-matched peers without reading deficits in a proportional reasoning task requiring them to compare rational (discrete or continuous) quantities and identify similarities among them. Previous studies have shown that while a late understanding of proportionality manifests when proportions consist of discrete sets, an early understanding of proportionality manifests when proportions consist of continuous amounts [35]. In other words, children from 6 years of age performed significantly worse when both the target and choice alternatives were represented with discrete quantities than when either or both of the proportions involved continuous quantities. This is probably because a discrete subdivision of whole quantities leads to counting procedures and overextension of numerical counts to proportional problems [48,49]. By applying the same paradigm to typical and dyslexic children, we hypothesize that the nature of the quantities included in the proportional problems differently influences children’s performance. Indeed, we expected to replicate Boyer et al.’s findings for what concerns typical children (good performance when both target and choice proportions are continuous, relatively poor performance when both target and choice proportions are discrete and at the chance level on mixed conditions because such stimuli contain discrete quantities). Conversely, since syllable segmentation might be difficult for dyslexic children, they may show difficulties in the discrete segmentation of a whole, therefore, we expected them to perform relatively well in the conditions where they can base their judgments on a discrete comparison but relatively poor in the condition where they have to segment a non-partitioned whole (continuous conditions). 

## 2. Materials and Methods

### 2.1. Participants

Thirty-two children were enrolled in the study, with half of the participants (n = 16) as controls and half of the participants (n = 16) with a formal evaluation of dyslexia. Within each subgroup, dyslexic and control children were matched for age (M ± SD 7–8 years old controls: 7.7 ± 0.4; M ± SD 7–8 years old dyslexic: 7.5 ± 0.5; M ± SD 9–10 years old controls: 9.4 ± 0.5; M ± SD 9–10 years old dyslexic: 9.6 ± 0.5) and sex (3/8 females in the 7–8 years old controls and 7–8 dyslexic subgroups: 4/8 females in the 9–10 years old controls and 9–10 dyslexic subgroups). Children were assigned to two age groups depending on their chronological age (7–8 years old; 9–10 years old). Age groups were identified based on the criterion for a formal diagnosis of dyslexia, which can be obtained only at the beginning of the third grade of the Italian primary school, which approximately corresponds to the age of 8 years old [50,51]. Therefore, analysis of children’s performance before 8 years of age can inform of methods to diagnose dyslexia with alternative tasks early, such as the one proposed in this study. The local ethics committee approved the study, and all children had written informed parental consent to participate. All methods were performed following the relevant guidelines and regulations. Dyslexic children were recruited from a private institute to assess and treat children with reading disabilities, while control children were recruited from local schools. 

### 2.2. Neuropsychological Assessment

The group of dyslexic children underwent a neuropsychological examination that comprised standardized tests typically administered to diagnose reading disabilities in primary school children: (1) DDE-2 Battery (Battery for the assessment of Developmental Dyslexia and Dysorthographia-2, Sartori & Job, 2007) to assess reading speed and accuracy (task 2 and 3) in word and non-word reading tasks; (2) MT battery (Prove di lettura MT per la scuola elementare-2, Cornoldi, Colpo, & Gruppo, 1998) to evaluate text reading speed and accuracy (two scores in total); (3) Wechsler Intelligence Scale for Children Fourth Edition (WISC-IV) (Wechsler, 2003) to evaluate general intelligence. The national criteria applied for a formal diagnosis of dyslexia is a poor performance on reading tasks in at least two out of six scores in two different reading tests (where poor performance is indicated as −2 standard deviations from the age-related norms) in the absence of poor intelligence scores. Children enrolled in both groups (dyslexic and control) reported intelligence scores within the age-matched norms (Intelligence Quotient—IQ, Verbal Comprehension Index—VCI, Processing Speed Index—PSI, Working Memory Index—WMI, Perceptual Reasoning Index—PRI). None of the children enrolled reported mathematical learning disabilities.

### 2.3. Procedure and Design

Mathematical thinking has been assessed with a paper and pencil adaptation of the task proposed by Boyer et al. [35], in which children were asked to provide judgments about proportionality without explicitly referring to the formal concept of proportions. Specifically, they were required to compare a target and a choice of alternative juice mixtures and select the choice juice whose proportions of fruit and water matched the target (Figure 1). Therefore, proportionality was determined by relative quantities of juice and water parts in each stimulus. Before the experiment started, participants were told a story about the teddy-bear Wally, who enjoys drinking all kinds of juices and likes to mix his juice himself. During the testing, instructions were verbally presented. On each trial, children were presented with the target juice and two alternative juices and were invited to help the character Wally identify the right mixtures for his different kinds of juice so that his juice tastes good like the target juice mixture. Therefore, they were required to point to one of the two choice alternative juice mixtures (right panel on each quadrant, Figure 1) that best represents the target juice mixture (left panel on each quadrant, Figure 1). Sixty-four self-paced trials were administered to each participant in four continuity conditions (four quadrants in Figure 1): each condition comprised sixteen trials randomly presented.

The difference between continuity conditions was in how the juice and water parts were represented. Indeed, target and choice alternatives could be represented with continuous water and juice amounts or with discrete water and juice units. In the discrete-discrete condition (DD), both the target and the choice juices were represented as blocks subdivided into a 1-cm unit. In the continuous-continuous condition (CC), both the target and the choice juices formed a unitary block not subdivided internally. In the discrete-continuous condition (DC), the target juice was presented as formed by the discrete units, while the choice juice was presented as a unitary block. In the continuous-discrete condition (CD), the target juice was presented as a unitary block, while the choice juice was presented as formed by the discrete units. Before starting the testing, all children were presented with four practice trials, one for each condition of the task, and were given feedback whenever required. No performance feedback was provided during the testing.

### 2.4. Analysis

On each trial, correct responses were registered as 1, and incorrect responses were registered as 0. For each participant, a proportional reasoning index has been calculated as the sum of correct/total trials, with the index ranging from 0 to 1. For each participant, five performance indexes were obtained: a Total index obtained by averaging all of the 64 trials administered, a DD index obtained by averaging only the trials in the DD condition (n = 16), a DC index obtained by averaging only the trials in the DC condition (n = 16), a CD index obtained by averaging only the trials in the CD condition (n = 16), a CC index obtained by averaging only the trials in the CC condition (n = 16). We evaluated the normal distribution of data applying the Shapiro–Wilk test of normality with the free software R (Free Software Foundation, Boston, MA, USA) and found that data did follow a normal distribution. A 2 × 2 ANOVA univariate-model analysis of variance was used to assess the performance across 64 trials with Total index as the dependent variable between age groups (7–8 vs. 9–10) and diagnosis (typical vs. dyslexic). Then, a 4 × 2 × 2 MANOVA multivariate-model analysis of variance was used to assess the performance calculated as partial scores (DD, CC, DC, CD) as dependent variables between age groups (7–8 vs. 9–10) and diagnosis (typical vs. dyslexic). Follow-up ANOVAs were performed to analyze the single effects of all the subscores separately. To assess whether performance at the proportional reasoning task correlates with a performance at the neuropsychological tests commonly used to diagnose dyslexia in Italy, we calculated the correlation coefficients between five score indexes (Total, DD, DC, CD, CC) and the six main scores from neuropsychological reading tests (reading word accuracy, reading word speed, reading non-word accuracy, reading non-word speed, reading text accuracy, reading text speed). For statistically significant correlation coefficients, we run a stepwise regression analysis to find the proportional reasoning score index that contributes most to predicting the reading performance of dyslexic children. 

## 3. Results

Our findings indicate that dyslexic children perform poorly in the proportional reasoning task compared to typical children independent of the age group considered, meaning that they are less accurate in performing judgments about proportionality compared to typical peers. This can be seen in Figure 2, which shows the Total index obtained as average correct responses across all trials (n = 64). 

We observed: (a) a significant main effect of age (F_1,28_ = 38.46, *p* < 0.001, partial η^2^ = 0.58) showing that children aged 7–8 years old performed poorer compared to children aged 9–10 years old (TOT_7–8 years old_ = 0.67 ± 0.25; TOT_9–10 years old_ = 0.84 ± 0.16); (b) a significant main effect of diagnosis (F_1,28_ = 139.77, *p* < 0.001, partial η^2^ = 0.83) showing that dyslexic children performed poorer compared to typical peers (TOT_dyslexic_ = 0.59 ± 0.17; TOT_typical =_ 0.91 ± 0.08); (c) a significant interaction age x diagnosis (F_1,28_ = 31.34, *p* < 0.001, partial η^2^ = 0.53) showing that the gap between dyslexic and typical performance is more evident in the younger group of children (TOT_dyslexic 7–8 years old_ = 0.43 ± 0.08; TOT_typical 7–8 years old_ = 0.91 ± 0.06; TOT_dyslexic 9–10 years old_ = 0.75 ± 0.04; TOT_typical 7–8 years old =_ 0.92 ± 0.1). The significant interaction age x diagnosis also indicates that while dyslexic children improve their performance from 7–8 to 9–10 years old (from 0.43 to 0.75), typical children do not show the same trend probably because their score is high already at the age of 7–8 years old (0.90). The performance of typical and dyslexic children has also been compared by taking into consideration the four partial subscores (DD, DC, CC, CD) resulting from the proportional reasoning task. Overall, the MANOVA indicated that performance is influenced by age (F_4,505_ = 19.1, *p* < 0.001; Wilks’ Λ = 0.87, partial η^2^ = 0.13) and diagnosis (F_4,505_ = 55.5, *p* < 0.001; Wilks’ Λ = 0.69, partial η^2^ = 0.3). Moreover, there was a statistically significant interaction effect between age and diagnosis on the combined dependent variables (F_4,505_ = 16.7, *p* < 0.001; Wilks’ Λ = 0.88, partial η^2^ = 0.12). To determine how dependent variables (DD, DC, CC, CD scores) differed for age and diagnosis, we performed follow-up ANOVAs (details can be found in Table 1). 

Overall, our analysis (a) confirmed previous results related to typical controls [35] by showing that younger children performed significantly worse than older children only when both the target and choice alternatives are represented with discrete quantities (subscore (t_221_ = −2.67, *p* < 0.01; DD_7–8 years old___typical_ = 0.82 ± 0.1, DD_9–10 years old_typical_ = 0.93 ± 0.09); (b) demonstrate that dyslexic children performed worse than typical peers independent of the fact that target and choice stimuli contain discrete or continuous quantities (DD_dyslexic_ = 0.57 ± 0.22, DD_typical_ = 0.87 ± 0.11; DC_dyslexic_ = 0.57 ± 0.20, DC_typical_ = 0.92 ± 0.09; CC_dyslexic_ = 0.64 ± 0.24, CC_typical_ = 0.93 ± 0.09; CD_dyslexic_ = 0.61 ± 0.24, CD_typical_ = 0.92 ± 0.11). A regression analysis was performed for both age groups of dyslexic children to assess whether the proportional reasoning task’s performance correlates with the performance at the neuropsychological tests commonly used to diagnose dyslexia. Results indicate that a significant correlation exists between reading accuracy and proportional reasoning performance in the DC condition for the younger group of dyslexic children (Figure 3). 

More specifically, the worse the performance at the DC condition of the proportional reasoning task, the worse the accuracy in reading words (r = 0.8, *p* < 0.01), non-word (r = 0.85, *p* < 0.01), and text (r = 0.73, *p* < 0.05). The stepwise regression analysis indicates that the best predictor of the performance at the DC condition of the proportional reasoning task is reading non-word accuracy (R = 0.79, R^2^ = 0.64, F_1,6_ = 10.4, *p* = 0.01). We did not find the same correlation in the 9–10 years old group of dyslexic children for the DC score, probably because our results indicate that at this age, the DC score of dyslexic and typical children is not significantly different (t_234_ = 2.06, *p* = 0.16).

## 4. Discussion

Numerous theories sought to explain the etiology of dyslexia. According to some evidence, dyslexia is associated with perceptual difficulties at the level of phonological representation [52,53], while different shreds of evidence indicate that dyslexia is linked with other perceptual and cognitive abilities, such as auditory temporal estimation [54,55], visual attentional span [56], as well as auditory and visual automatic orienting of spatial attention [57]. Moreover, some studies demonstrated that rhythmic perception [16,58,59,60] and auditory anticipation [55] are impaired in individuals with reading disabilities, suggesting a link between dyslexia and temporal perception. The idea that rhythm is linked with dyslexia has been supported by studies showing that rhythmical training can improve word reading and phonological awareness [61,62,63,64], probably because rhythm allows the segmentation of continuous speech into units [65,66]. Moreover, some studies hypothesized that speech and music share common aspects defined by the metrical organization of rhythm [13]. Overall, such works seem to indicate that the ability to metrically organize the material is important in order to develop good reading abilities, but no study to date has explored the link between reading impairments and other forms of thinking that require segmentation abilities, e.g., proportional reasoning. Proportional reasoning deals with the comparison of rational quantities, as musical rhythm deals with the comparison of temporal intervals, and speech perception deals with the sequential comparison of adjacent syllables. 

In the present study, we hypothesize that dyslexia could be linked to a more general impairment in processing the metrical structure of stimuli that might be independent of the domain considered (arithmetic, music, language). Therefore, we assessed the ability of dyslexic children to compare rational quantities in a child-friendly task and demonstrated that dyslexia is linked to an altered performance in a multiplicative (relative) form of thinking. Indeed, dyslexic children performed worse than typical peers in all the conditions of the proportional reasoning task administered, independent of the fact that target and choice stimuli contain discrete or continuous rational quantities. We might speculate that a proportional reasoning impairment could also affect reading because it hampers the ability to segment speech and language into smaller units, e.g., words in syllables, and vice versa. Such speculation would be supported by our finding that dyslexia and proportional reasoning are related in the DC condition of the task for the younger group (7–8 years old) of children. Indeed, the worse the accuracy in reading words, non-words, and text, the worse the performance in the DC condition of the proportional reasoning task. A possible interpretation is that the process of transforming a discrete quantity into a continuous quantity (what is requested for the DC condition) is similar to the process of reading a single word, which is made by consecutive “discrete” units (e.g., syllables). Therefore, we could expect that dyslexic children perform worse than sighted peers in this proportional reasoning because it is related to their difficulties in segmenting words. 

The present study shows some limitations and findings should be interpreted cautiously. One such limitation is that dyslexic children did not perform perceptual control tasks investigating their ability to segment auditory stimuli (e.g., musical intervals). Therefore, our results cannot be interpreted on the basis of a comprehensive perceptual partitioning deficit. This would have strengthened our hypothesis related to the common metrical structure of several domains related to reasoning, e.g., language and music. Moreover, the absence of control conditions (e.g., visual factors, language factors, and cognitive factors) hamper the generalization of our results to the population of dyslexic children. A second limitation is that since correct and incorrect responses present some perceptual similarity (e.g., stimulus size), it might be hypothesized that a more general deficit in control functions (e.g., executive functions) could explain the performance of dyslexic children. On the same line, a more parsimonious explanation of our results would advocate for a visual or attentional deficit of dyslexic children, causing them to misperceive the visual stimuli presented in the form of proportions. A further limitation is due to the small sample size enrolled, which is coherent with the preliminary power analysis but nonetheless may require caution in the interpretation of correlational results. Overall, our findings suggest that dyslexia is linked to a general form of reasoning based on a metrical structure, which would be the common feature on the basis of language, music, and mathematics. Several pieces of evidence have already shown that rhythm contributes to phonological awareness [61,62,63,64], probably due to the fact that rhythm facilitates the segmentation of a whole unit into smaller parts [65,66]. Therefore, we might speculate that reasoning based on meter facilitates reading skills because it permits the segmentation of words in syllables, similar to the influence of rhythm on speech perception. Moreover, the outcomes of this study suggest that dyslexia symptoms can be identified early by alternative non-reading tasks such as the proportional reasoning task used in this work, allowing its detection early in life before reading capabilities are acquired. In terms of social impact, early screening of reading difficulties could potentially anticipate neuropsychological interventions and prevent the risk of developing learning disabilities. Several studies used non-reading tasks (e.g., rhythmic tapping as in [67] or musical meter perception as in [13]) to predict reading development, but to our knowledge, no study to date has used a proportional reasoning task. Based on our findings, tasks based on proportional reasoning (e.g., mathematics) would allow us to predict reading impairments during school years and anticipate rehabilitation intervention.

## Figures and Tables

**Figure 1 brainsci-13-00795-f001:**
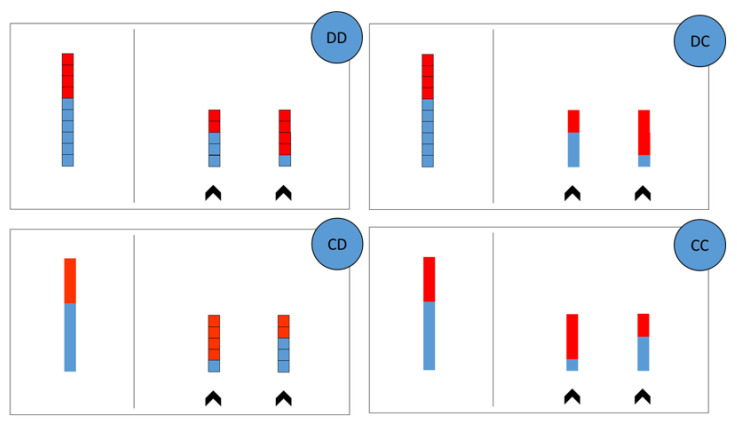
Proportional Reasoning Task. Continuity conditions of the proportional reasoning task adapted from Boyer et al. [19]. Conditions differ in how the target and the two choice alternatives are represented: DD (discrete-discrete), DC (discrete-continuous), CD (continuous-discrete), CC (continuous-continuous), here “discrete” implies blocks of 1-cm units and “continuous” implies no blocks.

**Figure 2 brainsci-13-00795-f002:**
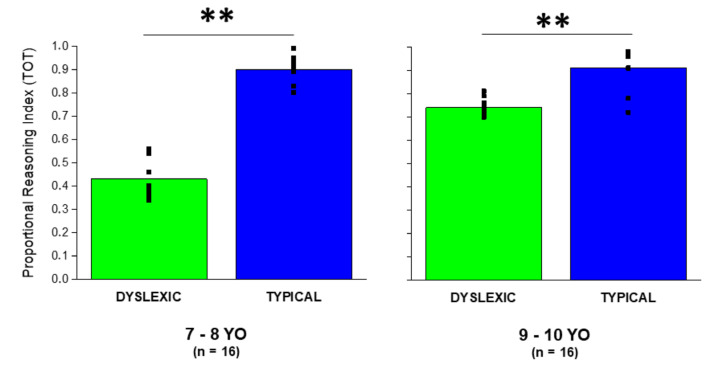
Comparison between typical and dyslexic children. Total index score is calculated as the average of correct responses across all trials (n = 64). The graphs clearly show that in both age ranges (7—8 and 9—10 years old), typical children outperformed dyslexic children when requested to provide judgments about proportionality. ** means *p* < 0.01.

**Figure 3 brainsci-13-00795-f003:**
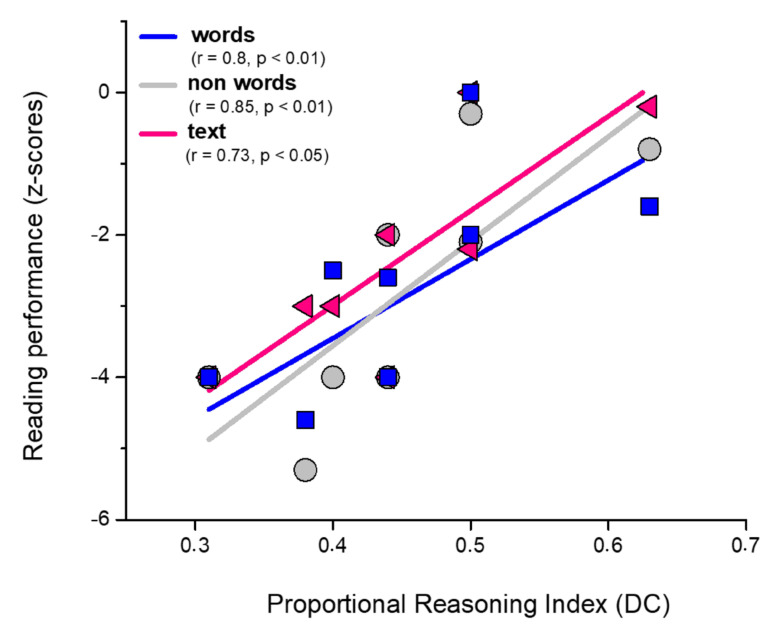
Correlation between reading and proportional reasoning abilities in dyslexic children. Correlation analysis indicates that a significant correlation exists between reading word (r = 0.8, *p* < 0.01), non-word (r = 0.85, *p* < 0.01), text (r = 0.73, *p* < 0.05) and proportional reasoning performance.

**Table 1 brainsci-13-00795-t001:** Follow-up ANOVAs results. In order to determine how dependent variables (DD, DC, CC, CD scores) differed for age and diagnosis, we performed a follow-up analysis, and the table shows the main descriptive, inferential statistics, and effect sizes between comparisons. * means statistical significance; ns means no statistical significance.

		F_1,508_	*p*	Partial η^2^	7–8 Years Old	9–10 Years Old
Main effect of age	DD	34.5	<0.001	0.06	0.62 ± 0.24	0.83 ± 0.18
	DC	9.8	<0.01	0.02	0.69 ± 0.26	0.8 ± 0.20
	CC	28.5	<0.001	0.05	0.7 ± 0.26	0.87 ± 0.15
	CD	44.9	<0.001	0.08	0.66 ± 0.29	0.87 ± 0.14
					controls	dyslexic
Main effect of diagnosis	DD	72.1	<0.001	0.12	0.87 ± 0.11	0.57 ± 0.22
	DC	96.8	<0.001	0.16	0.92 ± 0.09	0.57 ± 0.20
	CC	80.7	<0.001	0.14	0.93 ± 0.09	0.64 ± 0.24
	CD	97.3	<0.001	0.16	0.92 ± 0.11	0.61 ± 0.24
					controls	dyslexic
Interaction age × diagnosis	DD	8.01	<0.001	0.02	7–8 vs. 9–10 (*p* < 0.01 *)	7–8 vs. 9–10 (t_251_ = −5.3, *p* < 0.01)
	DC	12.8	<0.001	0.03	7–8 vs. 9–10 (ns)	7–8 vs. 9–10 (t_239_ = −8.1, *p* < 0.01)
	CC	26.01	<0.001	0.02	7–8 vs. 9–10 (ns)	7–8 vs. 9–10 (t_239_ = −5.6, *p* < 0.01)
	CD	51.6	<0.001	0.11	7–8 vs. 9–10 (ns)	7–8 vs. 9–10 (t_252_ = −3.8, *p* < 0.01)

## Data Availability

The raw data supporting the conclusions of this article will be made available by the authors, without undue reservation.

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
