# Peer review of "Proportional Reasoning Deficit in Dyslexia"

_brainsci, 2023, doi:10.3390/brainsci13050795_

Round 1
Reviewer 1 Report
Some citations are missing to support statements/arguments in the article. For example, in regards to "before the age of 8 years old children can be only categorized as “suspected dyslexia” 111 while after a formal diagnosis can be obtained", no evidence was provided.
No assumption tests were performed for ANOVA and MANOVA, and not clear what sort of regression/correlation analysis was performed.
16 dyslexic children's reading scores
Author Response
Response to Reviewer 1 Comments
Point 1: Some citations are missing to support statements/arguments in the article. For example, in regards to "before the age of 8 years old children can be only categorized as “suspected dyslexia” 111 while after a formal diagnosis can be obtained", no evidence was provided.
Response 1: Thank you for the comment. We provided references for the sentence indicate din the comment and we also checked and add othet citations throughout the text when missing.
- text at line 111: “Children were assigned to two age groups depending on their chronological age (7-8 years old; 9-10 years old). Age groups were identified based on the criterion for a formal diagnosis of dyslexia, which can be obtained only at the beginning of the third grade of the Italian primary school, which approximately corresponds to the age of 8 years old ([50, 51]”
Point 2: No assumption tests were performed for ANOVA and MANOVA, and not clear what sort of regression/correlation analysis was performed.
Response 2: Thank you for your comment. We actually performed assumption tests before proceeding with ANOVA and MANOVA but we did not report the results in the text. Specifically, we evaluated the normal distribution of data applying the Shapiro–Wilk test of normality with the free software R (Free Software Foundation, Boston, MA, United States) and found that data did follow a normal distribution, so we used parametric methods for the analysis. We provided more information about the statistical methods used in the Material and Methods section.
- text at line 181: “We evaluated the normal distribution of data applying the Shapiro–Wilk test of normality with the free software R (Free Software Foundation, Boston, MA, United States) and found that data did follow a normal distribution”
Other points:
(x) Extensive editing of English language and style required
Response 3: The paper was formally edited and revised by a native english speaker through an english-revision process provided by our institution.

Reviewer 2 Report
The paper presents a study that poses an objective of interest.
Increasing knowledge about the factors underlying dyslexia can be positive for a better and earlier identification, as well as to improve intervention.
However, the paper has some important limitations.
I present them below:
1. Results of neuropsychological evaluation.
Are there differences between the dyslexia group and the comparison group in the neuropsychological variables evaluated?
This may have an influence on the ability to solve the proportions tasks. If there are differences in IQ and the neuropsychological variables evaluated, this may be the cause of the differences in the ability to understand proportions.
2. Interpretation of the results.
I believe that the conclusions drawn in the paper cannot be drawn from the results obtained.
The results report differences between the two groups in scores on a proportion reasoning task.
A significant correlation is also reported between reading ability and ability to reason proportions.
And the results of a regression analysis are also reported (lines 250-255).
None of the statements indicated in the abstract in lines 10-15 can be inferred from these results.
Neither the correlations nor the comparisons between groups can lead to the conclusion that there is a causal or other relationship between dyslexia and reasoning of proportions.
Based on the results of the regression analysis, in any case it could be argued that some reading skills can be predictors of some skills in reasoning of proportions.
But this is not consistent with the sentence indicated in the abstract "These findings indicate that dyslexia can be identified early with alternative non-reading tasks such as the proportional reasoning task used in this work".
3. Sample size.
As the authors indicate, the sample size is small.
I think that the conclusions drawn from the results are too daring to have used such a small sample.
My assessment of the paper is that, although some results that may be of interest are presented, the conclusions drawn from these results are excessively ambitious.
Author Response
Response to Reviewer 2 Comments
Point 1: Results of neuropsychological evaluation. Are there differences between the dyslexia group and the comparison group in the neuropsychological variables evaluated? This may have an influence on the ability to solve the proportions tasks. If there are differences in IQ and the neuropsychological variables evaluated, this may be the cause of the differences in the ability to understand proportions.
Response 1: Thank you for your comment. As a premise (as reported at line 122), we have to say that the neuropsychological examination entailing the formal evalutation of reading impairments (e.g. DDE-2, MT) was performed only in dyslexic children, so we cannot compare the two groups (dyslexic and control) for such variables (e.g. reading speed and accuracy). However, as inclusion criterion we recruited control participants who did not show reading impairments as reported from the formal annual evaluation from teachers at school, so we can say that control participants did not present reading impairments. In this sense, we are sure that the two groups differ for what concerns reading abilities, which was indeed the core aim of the present work (to compare children with good and poor reading abilities in a proportional resoning task). For what concerns IQ measures, for both groups we recruited participants with IQ > 80, which is the cutoff for normal IQ performance. Therefore, in this case we are sure that both groups present normal intellectual skills and performance at the proportional reasoning task cannot depend on this variable. We changed the text accordingly in order to provide more information about the sample.
- text at line 133: “Children enrolled in both groups (dyslexic and control) reported intelligence scores within the age-matched norms (Intelligence Quotient - IQ, Verbal Comprehension Index - VCI, Processing Speed Index - PSI, Working Memory Index - WMI, Perceptual Reasoning Index - PRI). None of the children enrolled reported mathematical learning disabilities”
Point 2: Interpretation of the results. I believe that the conclusions drawn in the paper cannot be drawn from the results obtained. The results report differences between the two groups in scores on a proportion reasoning task. A significant correlation is also reported between reading ability and ability to reason proportions. And the results of a regression analysis are also reported (lines 250-255). None of the statements indicated in the abstract in lines 10-15 can be inferred from these results. Neither the correlations nor the comparisons between groups can lead to the conclusion that there is a causal or other relationship between dyslexia and reasoning of proportions. Based on the results of the regression analysis, in any case it could be argued that some reading skills can be predictors of some skills in reasoning of proportions. But this is not consistent with the sentence indicated in the abstract "These findings indicate that dyslexia can be identified early with alternative non-reading tasks such as the proportional reasoning task used in this work".
Response 2: Thank you for the comment. Indeed, we agree that our conclusions and interpretation might seem too daring and we modified the paper accordingly in the abstract.
- text at line 10-17: “In the present study, we assessed proportional reasoning in 16 dyslexic children and 16 age-matched controls from 7 to 10 years of age in order to investigate whether dyslexia might be also linked to an altered form of metrical thinking. We found that dyslexic children are less ac-curate in performing judgments about proportionality compared to typical peers and that read-ing accuracy correlates with proportional reasoning abilities for 7-8 year old children. Overall, these findings suggest that a link exists between reading and proportional reasoning abilities. We might speculate that fostering reasoning based on meter can facilitate reading because it permits to segment words in syllables and that dyslexia can be identified early with alternative non-reading tasks such as the proportional reasoning task used in this work.”
Point 3: Sample size. As the authors indicate, the sample size is small. I think that the conclusions drawn from the results are too daring to have used such a small sample. My assessment of the paper is that, although some results that may be of interest are presented, the conclusions drawn from these results are excessively ambitious.
Response 3: Thank you for the comment. Indeed, we agree that our conclusions and interpretation might seem too daring and we modified the paper accordingly, both in the abstract (see above) and in the discussion sections. Particularly, we specified that some results were interpreted in terms of speculations based on the link with other works related to the same topic.
Other points:
(x) English language and style are fine/minor spell check required
Response 4: The paper was formally edited and revised by a native english speaker through an english-revision process provided by our institution.

Reviewer 3 Report
These comments are slightly better presented in the attached pdf.
The article is original and pleasant to read.
1) However, in the form, you should:
(a) remove the repetition of "and" (p. 3, line 108)
(b) review the unusual use in developmental psychology of the abbreviation yo (for years old). If maintained, it should be made explicit when first used.
(c) homogenize the writing of decimal numbers: use a point (.) rather than a comma (,). Especially since, in the df column of Table 1 for example, the comma in 1,508 is a separator and not a decimal marker.
(d) remove the asterisk (*) in Table 1 because on the one hand you do not indicate its meaning, and on the other hand (as it probably means that the test is significant) you should also add asterisks to all the ps in the column labeled "p".
(e) delete the df column by replacing “F” with “F(1,508)”, or F1,508 (1,508 in index) as you write elsewhere, in the title of the preceding column.
(f) associate F(1,508) = 9.8 with p < .01 because you report ps < .01 in the same table leading one to believe that p is not less than .01 when it is actually less than .002.
2) On the substance, the argument is interesting but I would not say that the results support it "remarkably" well (p.8, line 284). Indeed, in addition to the limitations pointed out by the authors, one must also take into account that the multiplicity of statistical tests can lead to significant results by chance. Furthermore, the observed results were not predicted. On the contrary, it would have been predicted that it was in the DD condition that dyslexic students would have difficulties. Also, the differences between the DD, DC, CD and CC conditions appear to be small. Finally, the responses of the 7-8 year old dyslexic students do not deviate from the chance, and are even below it according to Figure 2, possibly not significantly.
Given the comorbidity of dyslexia and dyscalculia, this result was expected. Fischer (2007, Combien y a-t-il d'élèves dyscalculiques ? A.N.A.E., 19(3), 141-148), has shown that in the French national assessments, the correlation between French and Calculus is r = .670, and that it is even stronger when considering the correlation between French and mathematics deprived of calculus items (r = .744). Here, with proportionality reasoning, we are indeed in the presence of mathematics without calculation. Fischer's data therefore show that the results (total score or correlation between reading and proportional reasoning in dyslexic children) obtained are not really surprising because the correlation between language and a mathematics test without calculation is very strong. I hypothesize that it could have been obtained with another math test without calculus.
3) Moreover, statistically, having one result that is significant and another that is not does not imply that there is a significant difference between the two (see, for example, doi: 10.1007/s10339-016-0782-5).
In addition, I have some reason to worry about the statistical treatment because the only verifiable result—namely t(234) = 2.06, p = 0.16— is not consistent: with these degrees of freedom and value of the t-statistic, p reported by Excel is 0.041.
Conclusion. With minor corrections and form modifications, and perhaps the inclusion of the question of comorbidity in the interpretation, the article should be published.

Author Response

(The authors gave the same response as above.)

Round 2
Reviewer 2 Report
Unfortunately, I believe that the paper still has some limitations.
There are three main reasons:
1. In the current version it is explained that in the two groups the WISC scores were within the age-matched norms.
This information is important.
But even though both groups are in the average ranges for their age, it is possible that there are differences between the groups in intelligence.
If so, I think it is quite likely that these differences can explain a good part of the differences in the proportionality tasks.
As long as this issue is not resolved, I believe that the article cannot be published.
2. The following sentence is confusing: "The stepwise regression analysis indicates that the best predictor of the performance at the DC condition of the proportional reasoning task is reading non-word accuracy (R = 0.79, R2 = 0.64, F1,6 = 10.4 , p = 0.01)".
Does this mean that reading predicts the ability to solve proportionality tasks?
If so, this analysis does not seem adequate for the purpose of the paper.
Rather, what would need to be known is whether the ability to solve proportionality tasks at an early age may somehow indicate a later risk of reading difficulties.
3. Finally, in addition to these two issues that are strictly essential to publish the paper, I think there is another relevant issue: the theoretical arguments about how the ability to solve proportionality tasks can be related (if it is), with reading ability.
For this, it would be necessary, at least, to explain how the phonological reading route works in transparent languages ​​(such as Italian).
In addition, it would be necessary to justify whether or not these same results could be replicated in opaque languages ​​such as English, in which the preferred reading route is the visual route.